# Study on the Road Performance of Foamed Warm-Mixed Reclaimed Semi-Flexible Asphalt Pavement Material

**DOI:** 10.3390/ma14185379

**Published:** 2021-09-17

**Authors:** Jiawen Xie, Wenke Huang, Bei Hu, Zhicheng Xiao, Hafiz Muhammad Zahid Hassan, Kuanghuai Wu

**Affiliations:** School of Civil Engineering, Guangzhou University, Guangzhou 510006, China; xjw850843479@163.com (J.X.); 13026335960@163.com (B.H.); xzc297055783@163.com (Z.X.); zahid.hassan197@gmail.com (H.M.Z.H.)

**Keywords:** foamed warm-mixed reclaimed, semi-flexible pavement materials, road performance, grey correlation analysis

## Abstract

Warm-mixed reclaimed asphalt pavement (RAP) technology has been widely studied worldwide as a recycled environmental method to reuse waste materials. However, the aggregate skeleton structure of the warm-mixed reclaimed asphalt mixture is not stable because of the existence of the recycled materials. Warm-mixed recycled semi-flexible pavement material can solve the defects of the above materials. In this study, five different types of open-graded asphalt mixtures containing different contents of RAP were designed, and relevant laboratory tests were conducted to assess the road performance of the warm-mixed recycled semi-flexible pavement material. The test results indicated that the road performance of warm-mixed reclaimed semi-flexible pavement materials has good resistance to rut deformation ability. Furthermore, the materials also had good water stability and fatigue performance. The grey correlation analysis shows that the asphalt binder content has the most significant correlation with the high-temperature stability, and the correlation between RAP content and the fatigue performance was the greatest. Furthermore, the curing age has the most remarkable with the low-temperature crack resistance of the warm-mixed reclaimed semi-flexible material.

## 1. Introduction

Asphalt mixture has become one of the most widely used pavement materials globally because of its smooth surface, low noise and low maintenance cost. Aggregate plays a significant role in asphalt pavement, as aggregate accounts for approximately 95% of the weight and 90% of volume in asphalt mixtures. Asphalt binder becomes brittle after several years of use, and cracks appear due to oxidation and repeated wheel load [1]. The destruction of a large amount of asphalt mixture waste generated in road maintenance has caused severe pollution. Therefore, experts have repeatedly tried to use waste asphalt mixture [2,3,4,5]. RAP technology reduces the use of new aggregate material, allowing construction waste material in semi-flexible pavement construction. A reasonable amount of new aggregate material, new asphalt binder and waste asphalt material are mixed to form a new asphalt mixture with significant performance, according to the standard requirements. The recycled or waste asphalt is heated and mixed at a temperature of 170 °C in the laboratory, and the resulting recycled material shows significant performance. In addition, the heating of asphalt mixtures requires a considerable amount of energy, increases carbon dioxide in the environment, and produces toxic gases, which causes environmental effects [6,7,8,9].

Due to its advantages of low energy consumption, low emissions, and reducing the ageing of old asphalt binders, a new type of green and environmentally friendly asphalt recycled material-foam warm-mixed recycled asphalt mixture has been applied in recent years [10,11,12]. The foamed warm-mix asphalt technology aims to produce foamed asphalt binder by adding a specific amount of water to the foaming equipment. When water contacts the heated asphalt binder, it quickly evaporates and expands. The increased working capacity of asphalt enables it to be fully incorporated into aggregates at lower temperatures and to mix and compact the mixture at lower temperatures. However, the foamed warm-mix RAP technology can minimize the viscosity of the asphalt binder, resulting in a decrease in the compaction and mixing temperature. RAP technology has a slight advantage over hot-mix asphalt mixture in terms of low-temperature crack resistance, high-temperature performance and water stability.

Semi-flexible pavement combines certain best properties of flexible and rigid pavement by using an open-graded matrix asphalt mixture filled with specific cement grouting materials (void ratio 20–25%) [13]. Asphalt’s flexibility, seamlessness and water-resistance are complementary to high static load carrying capacity, resistance to rutting and abrasion, and resistance to oil and fuel leakage, which are also characteristics of typical concrete surfaces [14,15]. The cement grouting material can be interlocked with the basic asphalt mixture to improve the performance of the pavement as a semi-flexible pavement material. This research proposes a warm mixed recycled semi-flexible pavement material to improve the road performance of foamed warm mixed recycled asphalt mixture, which is driven by this unique semi-flexible mixed composite pavement type.

In this paper, the warm-mixed reclaimed semi-flexible pavement material was prepared based on the coarse aggregate void-filling method (CAVF). Five different types of open-graded asphalt mixtures containing different contents of RAP (0%, 15%, 30%, 45%, 60%, in mass), while the target air voids of 25% were designed. Relevant laboratory tests were conducted to evaluate the road performance of the materials, and the grey correlation degree of various factors on the road performance of warm-mixed reclaimed semi-flexible materials was analyzed.

## 2. Experimental Program

### 2.1. Raw Materials

Shell Road Solution Xinyue (Foshan, Guangdong Province, China) Co., Ltd. provided raw asphalt with a penetration grade of 60–80 (0.1 mm), which was used as the asphalt binder in this work. The technical qualities of the asphalt binder, including penetration, ductility, density and solubility, were tested following Chinese standard requirements (JTG E20-2011), and the obtained results are listed in Table 1. The coarse and fine aggregates were crushed diabase stones, while the mineral filler was limestone powder. Technical property tests, including water absorption, crushing, abrasion, and relative apparent density, were conducted following Chinese standard requirements (JTG E42-2005). Clay content and sand equivalent test for fine aggregate and relative apparent density and moisture content were performed on mineral powder. The fundamental properties of coarse aggregate, fine aggregate and mineral filler meet the Chinese standard requirements and are presented in Table 2.

The aggregate material used in this study was given by The Guangzhou Municipal Engineering and Maintenance Office Co., Ltd. in Guangzhou, Guangdong Province, China. All of the fundamental properties of coarse aggregate, fine aggregate and recycled material were tested in the laboratory according to the Chinese standards as mentioned in JTG F41-2008. The recycled material, which includes coarse aggregate, fine aggregate and asphalt binder, was dried and appropriately sieved. RAP materials with diameters less than 4.75 mm were removed in this study to guarantee that the effective void ratio of the asphalt mixture could meet the essential grouting criteria, as shown in Figure 1. According to Chinese requirements, the asphalt binder in RAP was removed using centrifugal separation and subsequently recovered using the Abson process (JTG F41-2008). Table 3 shows the aggregate and preserved asphalt binder parameters in detail.

A dry powder used in this study was given by Longhu Technology Co., Ltd. Shantou Guangdong, China. As indicated in Figure 2, the dry powder mortar contains cement, sand, and other ingredients. Table 4 shows the gradation of cement mortar.

### 2.2. Mix Design

The coarse aggregate void-filling method (CAVF) proposed in previous research [16,17] was applied to design the warm-mixed reclaimed matrix asphalt mixture.

The principle of this method is that the voids formed by coarse aggregate are filled with target porosity, fine aggregate, mineral filler and asphalt binder. Thus, the first step of this design procedure is to measure the voids fraction formed by the coarse aggregates in the tamping state. The voids in coarse aggregates (namely *VCA*) can be calculated with Equation (1) expressed as:(1)$$VCA=\left(1-\frac{\rho }{\rho_{b}}\right)\times 100$$
where *VCA* is the voids in coarse aggregate in the dry tamping test; ρ and $\rho_{b}$ are the packing density of the coarse aggregate (C.A) (g/cm^3^) and the apparent density of the C.A (g/cm^3^), respectively.

Then, the relationship between C.A, F.A, powder filler, asphalt binder and designed air void of the asphalt mixture can be expressed with Equations (2) and (3) shown as follows:(2)$$q_{c}+q_{f}+q_{p}=100\;$$
(3)$$\frac{q_{c}}{100\times \rho }\left(VCA-V_{V}\right)=\frac{q_{f}}{\rho_{af}}+\frac{q_{p}}{\rho_{f}}+\frac{q_{a}}{\rho_{a}}\;$$
where $q_{c}$, $q_{f}$, $q_{p}$ and $q_{a}$ are the mass percentage of C.A, mass percentage of fine aggregate, mass percentage of powder filler and mass percentage of asphalt binder, respectively. ρ, $\rho_{af}$, $\rho_{f}$, and $\rho_{a}$ are the packing density of the coarse aggregate (g/cm^3^), the apparent relative densities of fine aggregate, the density of powder filler and density of asphalt binder, respectively. *VCA* and $V_{V}$ are the voids in coarse aggregate in the dry tamping and the designed air void of the asphalt mixture, respectively.

In this study, five types of open-graded asphalt mixture containing different contents of RAP (0%, 15%, 30%, 45%, 60%, in mass) with the target air voids of 25% were designed. The gradations of the five asphalt mixture types were designed according to the CAVF method and the optimal asphalt contents were determined by the Marshall tests (ASTM D 1559) [18,19]. The asphalt binder in RAP was considered to contribute to the total asphalt content. Thus, the content of the foamed asphalt was determined by subtracting the asphalt binder content in RAP. Mixture ratios of the matrix asphalt mixture are shown in Table 5. 

### 2.3. Sample Preparation

#### 2.3.1. Preparation of Warm-Mixed Reclaimed Matrix Asphalt Mixture

This study applied the WLB10S asphalt foaming tester produced by Wirtgen Co., Ltd., Windhagen, Germany, as shown in Figure 1, to fabricate the matrix asphalt mixture. Previous research has shown that the foaming water content and foaming temperature play a leading role in the foaming effect at given water temperature, air, and water pressure [20]. To determine the appropriate heating temperature and foaming water content for better foaming effect of the asphalt binder, five different temperatures (145, 150, 155, 160, and 165 °C) and five foaming water contents (1%, 1.2%, 1.5%, 2%, and 2.5%, in mass) were selected. According to the Chinese specifications (JTG/T 5521-2019), the expansion ratio and the half-life time of the asphalt foaming should be more than ten times and 8 s, respectively. The foaming ability index defined as the area above the minimum expansion ratio on the expansion ratio-time fitting curve was selected as the foaming criterion. A more extensive foaming ability index represents a better foaming effect of the asphalt binder.

The expansion rate and half-life of the asphalt foaming with five foaming water contents at different temperatures are presented in Figure 3. As shown from Figure 3, the expansion ratio increases with the increase of foaming water content, while the half-life time decreases with the increase of foaming water content. The technical requirements of foamed asphalt should meet the expansion ratio of not less than 10 times and half-life time of not less than 8 s, which means that 1.5% of foaming water content at different temperature meet the specification requirements (the expansion ratio and half-life time at a constant foaming water content totally above the dotted line, shown in Figure 3). The foaming ability index of the asphalt binder at different temperatures was shown in Figure 4. As can be seen from Figure 4, the foaming ability index reaches the highest point with 1.5% of foaming water content at temperature 155 °C. Thus, the optimal foaming temperature and foaming water content for the foaming asphalt binder used in this study should be 155 °C and 1.5%, respectively.

#### 2.3.2. Preparation of Cement Mortar

Cement mortar needs to meet certain fluidity and mechanical strength to ensure the performance of semi-flexible pavement materials referring to Guangdong provincial, a local standard for cement mortar (DB44/T1296-2014) as shown in Table 6. The only variable was the water–cement ratio in cement mortar when the gradation of dry powder mortar was given. This study measured the performance indices, including fluidity, compressive strength, flexural strength, and dry shrinkage to determine the best water–cement ratio for cement mortar.

The water–cement ratio selected in this study was 0.20, 0.21, 0.22, 0.23 and 0.24. The fluidity of cement mortar with a water–cement ratio higher than 0.21 can meet the requirement as shown in Table 7. The test results of compressive strength, flexural strength and dry shrinkage for 7 and 28 days of curing time are presented in Table 8. Shown in Table 8, with the increase of water–cement ratio, the flexural strength and compressive strength of cement mortar decrease, while the dry shrinkage increases under the same curing time. However, the flexural and compressive strength for cement mortar with a water–cement ratio of 0.24 cannot meet the requirements in 28 days of curing. Properties of cement mortar with a water–cement ratio of 0.22 and 0.23 can be meet the technical requirements. However, the cement mortar properties with a water–cement ratio of 0.22 are slightly better than that of 0.23. Thus, the optimal water–cement ratio was determined to be 0.22 in this study.

#### 2.3.3. Preparation of Warm-Mixed Reclaimed Semi-Flexible Mixture

The Marshall test method was conducted to prepare the specimens in this study. The detailed step-by-step procedure for preparing the testing samples are presented as follows:(1)The asphalt mixture was mixed and prepared at 120 °C according to the designed mixtures ratios of asphalt mixtures as given in Table 5 [21]. The dimensions of the prepared specimens were 450 × 150 ×185 mm and 320 × 260 × 60 mm for small slabs and 101.5 mm in diameter and 63.5 mm in height for cylindrical samples, which were prepared by Marshal method. All of the prepared samples were kept for drying for 24 h at normal room temperature and demolded after 24 h;(2)A cement mortar was prepared according to the design gradation of powder mortar as presented in Table 4. The water–cement ratio and sand content of 0.22 and 30.25% were used to prepare cement mortar, respectively. The edges and bottom of the prepared specimens were covered with transparent sheets to avoid leaking while preparing samples in the laboratory;(3)The mold was filled with the cement mortar and all the bubbles on the surface were removed with the vibration table. The extra cement mortar from the surface of specimen was removed by using a scraper after filling while preparation in the laboratory;(4)The prepared specimens with cement mortar were placed for 24 h at normal room temperature. After proper drying, the transparent sheets were removed, and the specimens were placed in a curing box having 20 ± 3 °C temperature. The humidity of the curing box was more than 80% for proper curing;(5)The prepared slab was cut into small beams to perform the fatigue and fracture tests by using the cutting machine in the laboratory. The dimensions of small beams were 380 × 50 × 63 mm and 250 × 30 × 35 mm for the four-point bending test and low-temperature fracture test, respectively;(6)The prepared cylindrical samples after proper curing and drying, the top surface and cross-section of the cylindrical samples with cement mortar are presented in Figure 5a–c. A cross-section of the samples shows that the aggregates are properly connected and have low air voids, as shown in Figure 5. The results indicate that the design mortar meets the standard requirements and has good fluidity.

## 3. Testing Program

### 3.1. High-Temperature Resistant Test

Rutting tests were conducted to investigate the high-temperature performance of the warm-mixed reclaimed semi-flexible pavement materials by the Chinese specifications (JTG E20-2011). According to the preparation procedure in Section 2.3.3, 320 × 260 × 60 mm asphalt mixture slabs with different contents of RAP materials (referring to Table 5) were fabricated for rutting tests, and then placed in a curing box with a temperature of 20 ± 3 °C and humidity of more than 80% to cure for 3 and 7 d. 

The dynamic rutting stability (DS) index in the rutting test calculated with Equation (4) was obtained through the Hamburg Wheel Tracking (HWT) test apparatus (Infera Test Pruftechnik GmbH, Backenheim, Germany). For each type of asphalt mixture, three replicate tests were performed.
(4)$$DS=\frac{\left(t_{2}-t_{1}\right)\times N}{d_{2}-d_{1}}\times c_{1}\times c_{2}$$
where $d_{1}$ and $d_{2}$ (mm) are tracking depths at 45 and 60 min, respectively. $t_{1}$ and $t_{2}$ represent 45 and 60 min, respectively. *N* is the rolling speed of the steel wheel which is 52 times/min in this study. $c_{1}$ and $c_{2}$ are correction factors that are equal to 1.0.

### 3.2. Low-Temperature Test

A three-point bending test evaluated the crack resistance of the prepared specimens with the cement mortar according to the Chinese standard test methods of asphalt and asphalt mixtures for highway engineering (JTG E20-2011, T 0715). The prepared slabs with cement mortar were placed for curing for three days and seven days and cut by using a cutting machine. The small beams’ dimensions for the three-point bending test were 250 × 30 × 35 mm in length, width, and height, respectively. The small cut beams were placed in the water bath at −10 °C temperature for six hours to perform the low-temperatures crack resistance test. The test was conducted at a loading rate of 50 mm/min. The flexural tensile strength, stiffness modulus and maximum flexural strain were used to evaluate the crack resistance, which can be calculated with Equations (5) and (6).
(5)$$R_{B}=\frac{3\times L\times P_{B}}{2\times b\times h^{2}}$$
(6)$$\varepsilon_{B}=\frac{6\times h\times d}{L^{2}}$$
where *L* is span length (mm); *h* is mid-span height (mm); *b* is mid-span width (mm); *d* is mid-span deflection at failure (mm); *P_B_* is the maximum load at failure (N); *R_B_* is the flexural tensile strength (MPa); *ε_B_* is the failure strain (με, 1 με = 10^−6^ε).

### 3.3. Moisture Damage Resistance Test

The moisture damage resistance performance of the warm-mixed reclaimed semi-flexible pavement materials was determined by the Marshall immersion test and the freeze–thaw splitting test. In the immersion Marshall test, the specimens were immersed in 60 ℃ water bath for 30 min and 48 h, and the stability of the specimens was measured, respectively. The immersion residual Marshall stability (MS) of the specimens can be calculated with Equation (7) expressed as:(7)$$MS=\frac{MS_{1}}{MS_{2}}\times 100$$
where MS is residual stability of the specimen (%). $MS_{2}$ and $MS_{1}$ (kN) are the stability of the specimen after immersion for 30 min and 48 h, respectively.

In the freeze–thaw splitting test (FTST), the prepared cylindrical specimens were divided into two groups. The first group was immersed in 25 °C water bath for 2 h, and then the indirect tensile strength (ITS) of the specimens was performed with a loading rate of 50 mm/min. The second group was immersed in a vacuum at room temperature for 15 min until it achieved at vacuum saturation, then placed in a freezer at a temperature of −18 °C for 16 h. Afterward, the specimens were placed into a 60 °C water bath for 24 h followed by placed into a 25 °C water bath for 2 h before testing. Finally, the indirect tensile strength of the second group was performed at 25 °C with a loading rate of 50 mm/min. The ITS ratio of the specimen before and after water conditioning was used to evaluate moisture damage resistance performance of the warm-mixed reclaimed semi-flexible pavement materials with Equation (8) expressed as:(8)$$TSR=\frac{R_{t1}}{R_{t2}}\times 100$$
where TSR is the indirect tensile strength ratio (%). $R_{t1}$ and $R_{t2}$ are the splitting strength of the specimens with and without freeze–thaw process (MPa).

### 3.4. Fatigue Test

Four-point bending test (4 PB) was conducted by Cooper NU-14 (Cooper Research Technology, Ripley, UK) to evaluate the fatigue performance of the warm-mixed reclaimed semi-flexible pavement materials under repeated load. The fatigue specimens with a dimension of 380 × 63.5 × 50 mm in length, width, and height, respectively, were cut from prepared slab specimens after curing for 28 days. The strain magnitudes selected in this study were 150 and 200 με. All the 4 PB tests were performed at 15 °C with a loading frequency of 10 Hz. The fatigue life of the specimens was defined as the number of loading cycles when the stiffness modulus of the specimen beam decreases to 50% of its initial stiffness modulus.

### 3.5. Grey Correlation Analysis (GCA)

J. Deng developed the grey correlation analysis (GCA) based on the development trend degree of similarity or dissimilarity between the factors to evaluate and optimize schemes with a multi-index [22]. The basic concept of this analysis method is to calculate the correlation coefficient and the correlation degree, and then evaluate the influence degree of different affecting factors by the reference sequences [23]. The evaluation process of grey correlation analysis are as follows:(1)Determination of reference sequence

The factors that affect the behavior of the system are composed of comparative sequences, which can be expressed as $x_{i}=\left\{x_{i}\left(k\right),k=1,2\ldots m,\;i=1,2\ldots n\right\}$. The factors that reflect the characteristics of the behavior of the system are composed of reference sequences, which can be expressed as $x_{0}=\left\{x_{0}\left(k\right),k=1,2\ldots m\right\}$;
(2)Normalize and get non-dimension

In this paper, the initial value converting method was used by dividing all the data of a sequence by its first value to normalize the values expressed as $X_{i}=\left\{x_{i}\left(k\right)/x_{i}\left(1\right),k=1,2\ldots m,\;i=1,2\ldots n\right\}$ and $X_{0}=\left\{x_{0}\left(k\right)/x_{0}\left(1\right),k=1,2\ldots m\right\}$;
(3)Calculation of grey correlation coefficient

Let the difference between the reference sequence and each comparative sequence set as $\Delta_{i}\left(k\right),\;\Delta_{i}\left(k\right)=\left|X_{0}-X_{i}\right|,\;i=1,2\ldots n,\;k=1,2\ldots m$. The maximum difference ${max}_{i}\;{max}_{k}\Delta_{i}\left(k\right)$, the minimum difference ${min}_{i}\;{min}_{k}\Delta_{i}\left(k\right)$ and the grey correlation coefficient $\xi_{i}\left(k\right)$ were calculated by Equations (9)–(11), respectively.
(9)$${max}_{i}\;{max}_{k}\Delta_{i}\left(k\right)=max_{i}\left(max_{k}\left|X_{0}\left(k\right)-X_{i}\left(k\right)\right|\right)$$
(10)$${min}_{i}\;{min}_{k}\Delta_{i}\left(k\right)=minx_{i}\left(min_{k}\left|X_{0}\left(k\right)-X_{i}\left(k\right)\right|\right)$$
(11)$$\xi_{i}\left(k\right)=\frac{{min}_{i}\;{min}_{k}\Delta_{i}\left(k\right)+\rho {max}_{i}\;{max}_{k}\Delta_{i}\left(k\right)}{\Delta_{i}\left(k\right)+\rho {max}_{i}\;{max}_{k}\Delta_{i}\left(k\right)}$$
where ρ is a coefficient, ρ=0.5 in this study;
(4)Determination of correlation degree

The correlation degree among the system factors was defined as $r_{i},r_{i}=\frac{1}{n}\overset{n}{\underset{k=1}{\sum }}\xi_{i}\left(k\right)\;,\;i=1,2\ldots n,\;k=1,2\ldots m$. 

## 4. Results and Discussion

### 4.1. Rutting Resistance

The rutting depth of the warm-mixed reclaimed semi-flexible pavement material at different curing ages are presented in Figure 6a,b. As shown in Figure 6, the rutting depth increases with the increase of RAP content in the same curing age, and the rutting depth of the material with the same content of RAP decreases with the increase of curing age. Compared with semi-flexible pavement without RAP, the maximum rutting depths of the material with a RAP content of 60% at 20,000 rolling cycles for 3 and 7 d curing age are 0.69 and 0.65 mm, respectively. This indicates that the warm-mixed reclaimed semi-flexible pavement material has high rutting resistance performance.

The influences of RAP contents and curing ages on DS are illustrated in Figure 7a,b. As it can be seen, the DS values decreased with the increase of RAP contents in the same curing age. To quantify the influence of RAP content on DS, the reduction ratio of DS calculated with Equation (12) is defined as a fraction of the reference DS value (DS value of the material with a RAP content of 0%). This is taken as a variation in DS value with different RAP content as shown in Figure 7a,b.
(12)$$reduction\;ratio\;of\;DS=\frac{variarion\;of\;DS\;value}{DS\;value\;of\;the\;material\;without\;RAP}\;$$

Figure 7 shows that the dosage increase for RAP is nonlinear to the reduction ratio of dynamic stability. The reduction ratio of dynamic stability for the material with RAP content increases from 0% to 15% develops faster than when the RAP content is from 15% to 60%. Raising the RAP content to 60%, the reduction ratio of dynamic stability for the material can reach to 50% and 60% in 3 and 7 curing days, respectively. In other words, the high content of RAP greatly influences the dynamic stability of the material. 

To clarify the influence of curing time on the DS, ratio of dynamic stability variation versus RAP content has been drawn, as displayed in Figure 8. It can be seen from Figure 8 that the DS variation of 7 curing days for the material without RAP increases by 50% compared with that of 3 curing days. The dynamic stability variation from 3 curing days to 7 curing days decreases with the increase of RAP content. When the RAP content is from 30% to 60%, the dynamic stability variation increase of dynamic stability is within 20%. This indicates that the increase of curing days has a slow effect on the DS of the material with a relatively high RAP content. This is mainly because the increase of curing day cannot counteract the decrease of dynamic stability caused by the increase of RAP content.

### 4.2. Low-Temperature Crack Resistance

The flexural tensile strength and failure strain of the warm-mixed reclaimed semi-flexible pavement material at different curing ages are presented in Figure 9. As shown in Figure 9, when the RAP contents range from 0% to 30%, the flexural tensile strength and failure strain decrease with the increase of RAP content. While the flexural tensile strength and failure strain change trends when the RAP content is from 30% to 60%. The main reason for this trend is that the reclaimed pavement materials contain old asphalt binder, forming weakened interfaces between old aggregates and asphalt mastic. The weakened interfaces lead to lower bond strength and deteriorate flexural tensile strength and failure strain of the warm-mixed, semi-flexible pavement material. However, the effective porosity of the matrix asphalt mixture will decrease when the reclaimed material content increases to 30–60%, according to our experimental results. Thus, the matrix asphalt mixture containing higher reclaimed material content becomes denser. The flexural tensile strength and failure strain of the warm-mixed reclaimed semi-flexible pavement material increases slightly with the increase of reclaimed materials.

### 4.3. Moisture Damage Resistance

Results of Marshall immersion tests and freeze–thaw splitting tests are presented in Figure 10. As shown from Figure 10, the immersion residual Marshall stability (MS) decreases with the increase of RAP content and increases with the rise of the curing ages. It can also be found that the values of MS are all above 100%, which indicates that the Marshall stability of 48 h immersion is more remarkable than of 30 min immersion. This is because a small portion of cement mortar in the semi-flexible material is not entirely hardened during the early curing age. When the Marshall specimen was placed in the constant temperature water bath box at 60 °C for 48 h, the strength of the semi-flexible material improved due to the hydration of the cement mortar. It can also be seen from Figure 10 that the indirect tensile strength ratio (*TSR*) of the specimens has the same trend in the increase of RAP content as Marshall stability (*MS*). The indirect tensile strength ratio values are above 90% when the curing age is up to 7 d, indicating that warm-mixed reclaimed semi-flexible pavement material has good water stability.

### 4.4. Fatigue Performance

The fatigue performance of the warm-mixed reclaimed semi-flexible pavement material is shown in Figure 11. As demonstrated from Figure 11, the fatigue life of the warm-mixed recycled semi-flexible material increases with the increase of RAP content at the same strain magnitude. At the same time, the initial stiffness modulus decreases with the increase of reclaimed materials. When the RAP content is the same, the fatigue life of the semi-flexible material decreases with the increase of strain level.

### 4.5. Grey Correlation Analysis

The comparative sequences of the factors affecting the system behavior, such as RAP content, asphalt binder content, curing age and strain magnitude, are listed as $\;x_{i}=\left\{x_{1},x_{2},x_{3},x_{4}\right\}$. The factors that reflect the characteristics of the behavior of the system, such as DS, Flexural tensile strength, Failure strain, MS, TRS and Fatigue life, are expressed as $x_{0i}=\left\{x_{01},x_{02},x_{03},x_{04},x_{05}\right\}$. The comparative sequences and reference sequences are shown in Table 9 and Table 10, respectively. The comparison sequences and reference sequences were dimensionless and treated by the method of initial value reduction, and then the grey correlation coefficients were calculated by Equations (10) and (12) in Section 3.5. Finally, the grey correlation degree of various factors on the road performance of warm-mixed reclaimed semi-flexible materials was obtained, as shown in Table 11.

As can be seen from Table 11, the order of the influence degree of each influencing factor on the high-temperature performance of warm-mixed reclaimed semi-flexible materials is: asphalt binder content > curing age > RAP content. This indicates that the asphalt binder content has the most significant correlation with the high-temperature stability of the warm-mixed reclaimed semi-flexible material. This is because the high asphalt content of the matrix asphalt mixture leads to the decrease of the porosity of the matrix asphalt mixture and the grouting amount of cement mortar becomes smaller, while the rutting resistance is mainly provided by cement mortar.

Furthermore, the order of the influence degree of each influencing factor on the material’s low-temperature crack resistance is: curing age > RAP content > asphalt binder content. This indicates that the curing age has the most significant with the low-temperature crack resistance of the warm-mixed reclaimed semi-flexible material. Cement mortar in this material is a rigid material with weak flexural strain resistance and is vulnerable to brittle failure. Thus, the curing age dominates the strength of cement mortar and greatly affects the low-temperature crack resistance of the material. The order of the influence degree of each influencing factor on the water stability performance of the material is as follows: curing age > asphalt binder content > RAP content. The results show that the curing age has the most remarkable correlation with the water stability of the warm-mixed reclaimed semi-flexible material. The order of the influence degree of each influencing factor on the fatigue property of the material is as follows: RAP content > strain magnitude > asphalt binder content. The correlation between RAP content and the fatigue performance of the semi-flexible materials is the greatest. The addition of used materials can significantly improve the fatigue life of the materials.

This study has analyzed the grey correlation degree of various factors on the road performance of warm-mixed reclaimed semi-flexible materials. It is found that multiple factors have different influence degree of the road performance of warm-mixed recycled semi-flexible materials and these factors restrict each other. The road performance of the material cannot be changed by controlling the single variable. It is necessary to consider the various factors to select a suitable variable range and choose several factors with more significant influence for research.

For example, the grey correlation degree of RAP content on the high-temperature stability, low-temperature crack resistance and water stability performance of the recycled semi-flexible material is low, but it cannot directly explain that the RAP content has little effect on the road performance of the material. According to the road performance test results, the change of RAP content has a more significant impact on the comprehensive path performance of the mixed regenerative semi-flexible material. Because there are correlations between the different factors, RAP content directly causes the change of other factors, and influences material road performance. In grey correlation analysis, selecting similar factors and comparing them with test results can improve the accuracy of results.

## 5. Conclusions

In this paper, a warm-mixed recycled semi-flexible pavement material was recommended to improve the road performances of the foamed warm-mixed RAP mixture. The road performances of the pavement material were investigated, and the grey correlation analysis was used to evaluate the influence degree of different affecting factors on the road performance of warm-mixed reclaimed semi-flexible materials. The following conclusions were drawn:(1)The rutting depth increases with the increase of RAP content in the same curing age. The reduction ratio of dynamic stability for the material with RAP content increases from 0% to 15% develops faster than when the RAP content is from 15% to 60%. When the RAP contents range from 0 to 30%, the flexural tensile strength and failure strain decrease with the increase of RAP content. The flexural tensile strength and failure strain changed trend when the RAP content is from 30% to 60%;(2)The immersion residual Marshall stability (MS) decreased with the increase of RAP content and increased with an increase of the curing ages. The freeze–thaw splitting strength ratio of these materials increased with the increase of curing age, the fatigue life of the warm-mixed reclaimed semi-flexible material increased with the increase of RAP content at the same strain magnitude. (3)The asphalt content has the most significant influence on the high-temperature stability of the material. The correlation between RAP content and the fatigue performance of the semi-flexible materials was the greatest. The curing age was the first in the order of the influence of various factors on the low-temperature crack resistance and water stability of warm-mixed reclaimed semi-flexible materials. 

## Figures and Tables

**Figure 1 materials-14-05379-f001:**
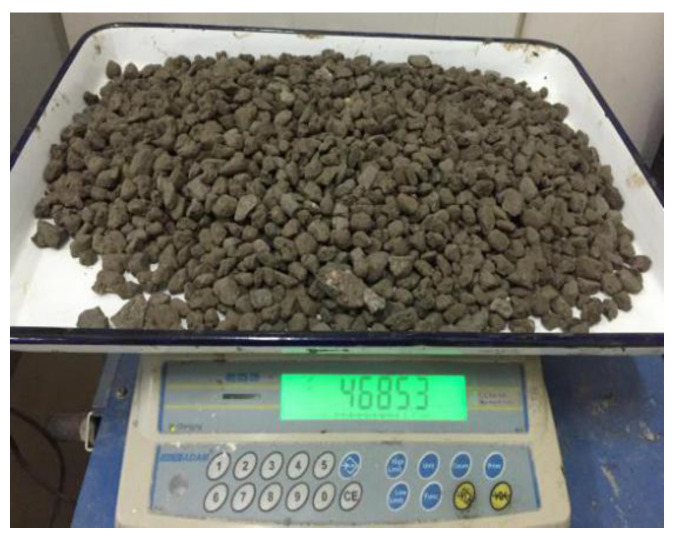
RAP material after sieving and drying.

**Figure 2 materials-14-05379-f002:**
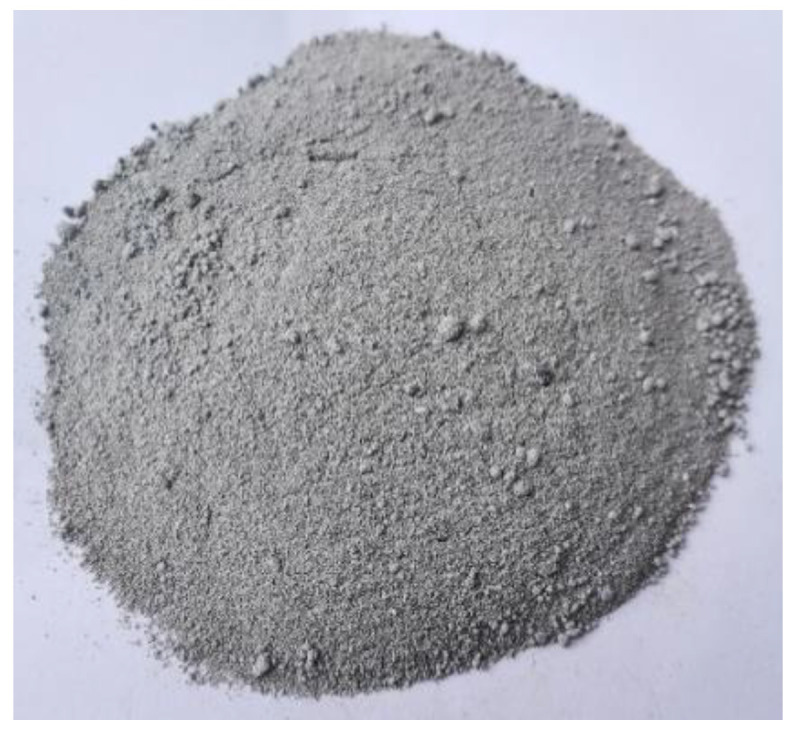
The dry powder mortar.

**Figure 3 materials-14-05379-f003:**
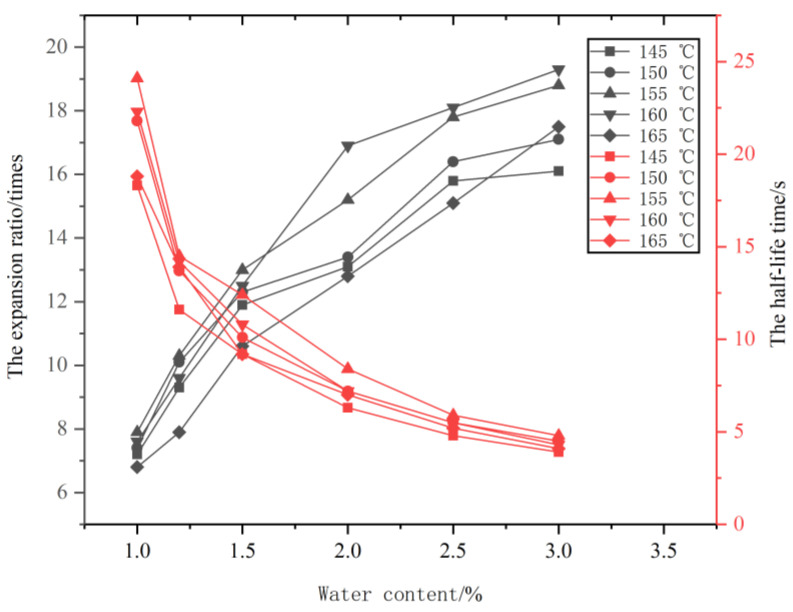
Expansion rate and half-life at different temperature.

**Figure 4 materials-14-05379-f004:**
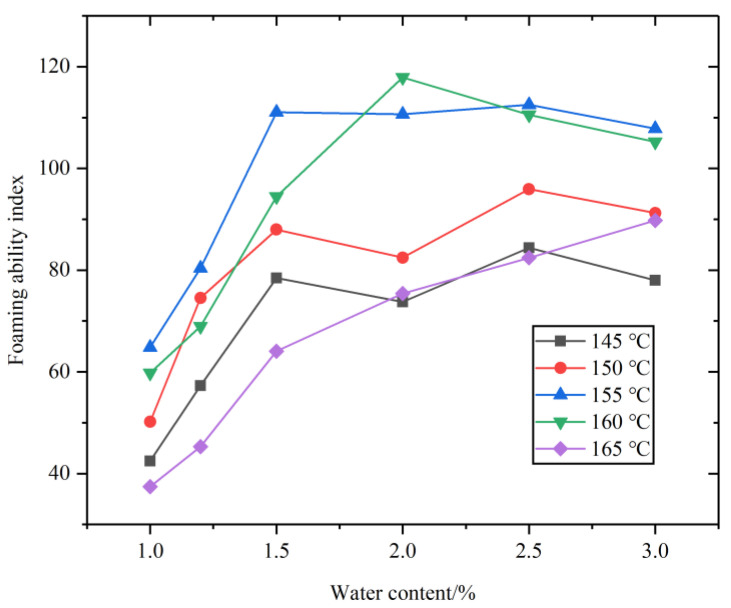
Foaming ability index at different temperature.

**Figure 5 materials-14-05379-f005:**
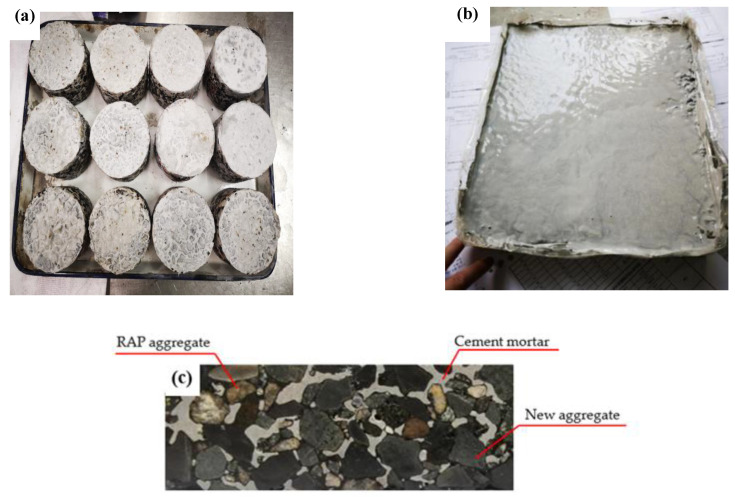
Prepared specimens. (**a**) Cylindrical samples filled with cement mortar; (**b**) slab samples filled with cement mortar; (**c**) cross-section of the warm-mixed reclaimed semi-flexible material.

**Figure 6 materials-14-05379-f006:**
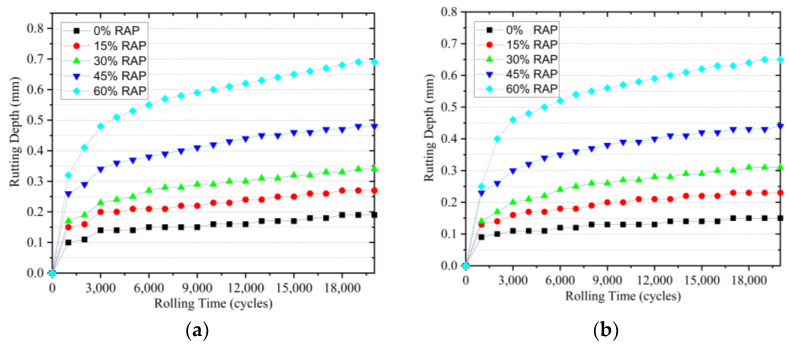
Rutting depth of the mixture at different curing age: (**a**) 3 days; (**b**) 7 days.

**Figure 7 materials-14-05379-f007:**
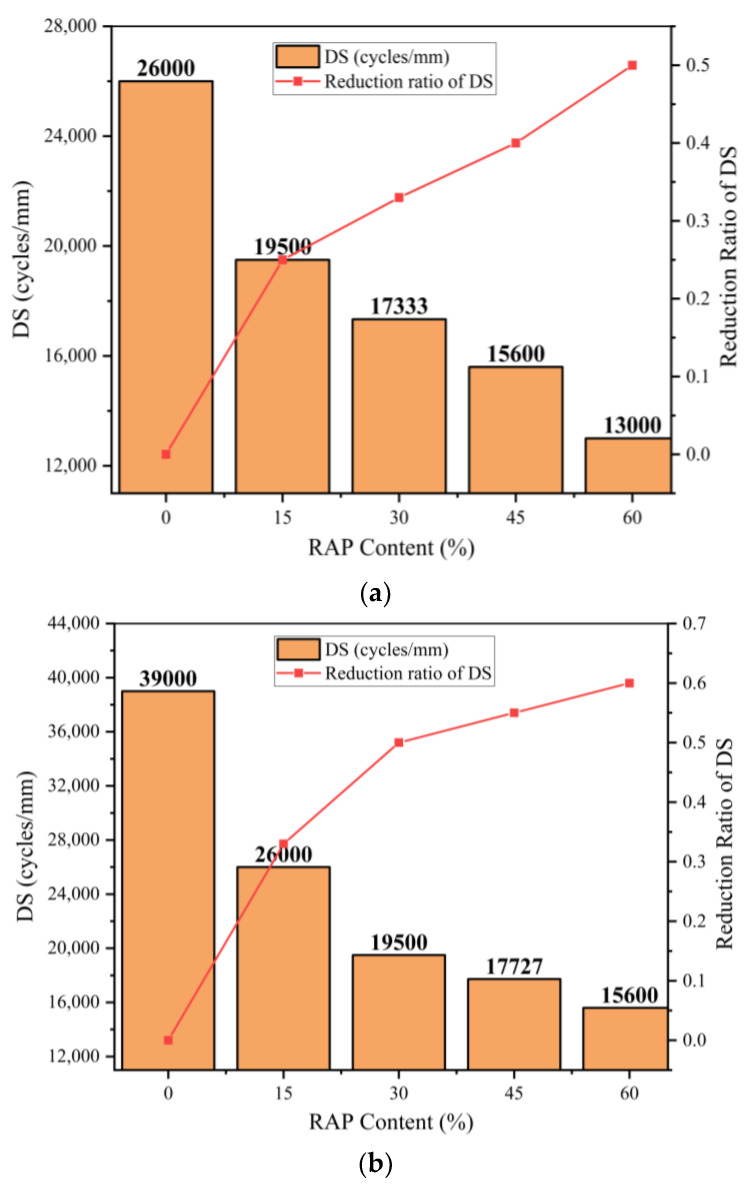
DS of the mixture at different curing age: (**a**) 3 days; (**b**) 7 days.

**Figure 8 materials-14-05379-f008:**
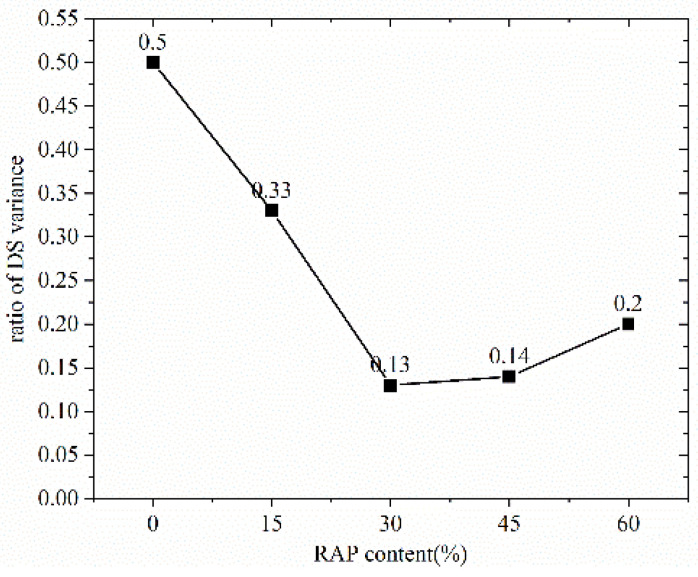
Ratio of DS variance with different RAP content.

**Figure 9 materials-14-05379-f009:**
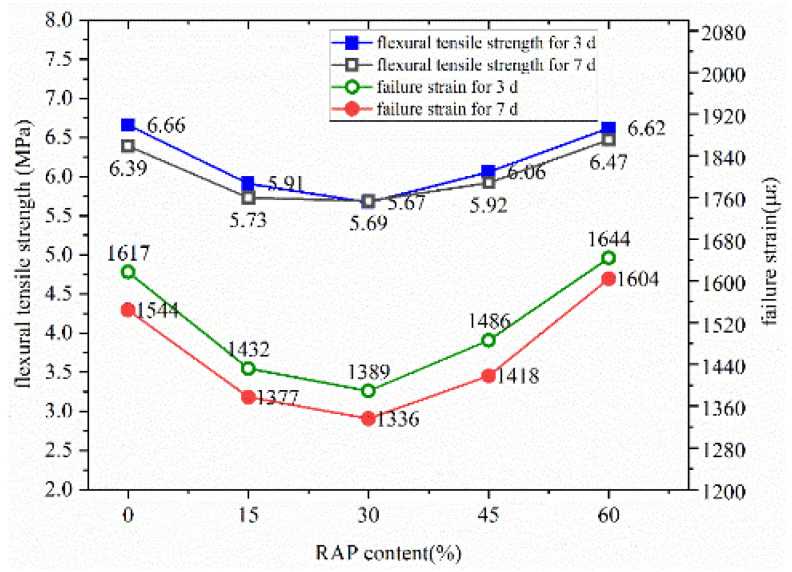
Flexural tensile strength and failure strain of the warm-mixed reclaimed semi-flexible pavement material at different curing ages.

**Figure 10 materials-14-05379-f010:**
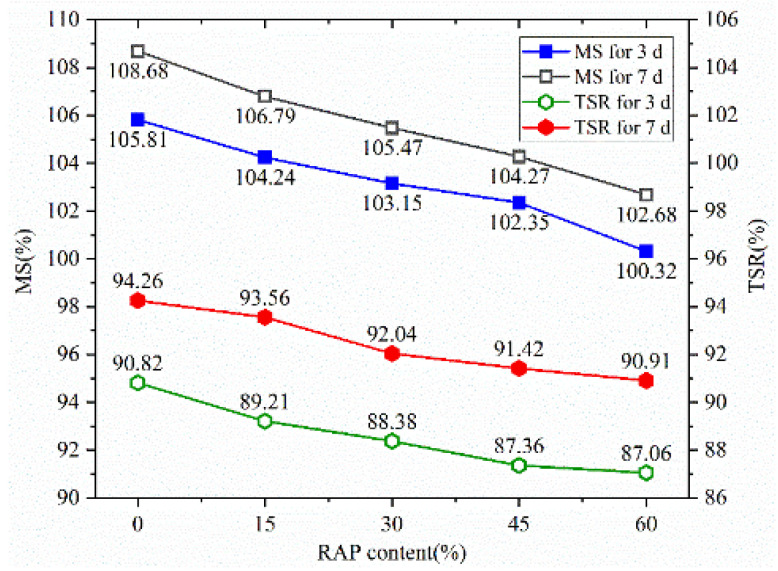
MS and TSR of the mixture at different curing ages.

**Figure 11 materials-14-05379-f011:**
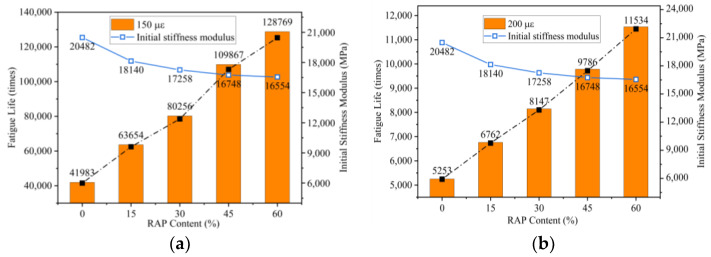
Fatigue life of the mixture at different strain magnitude:(**a**) 150 με; (**b**) 200 με.

**Table 1 materials-14-05379-t001:** Basic properties of asphalt binder.

Properties	Unit	Test Results	Test Method
Penetration at 25 °C	0.1 mm	71.8	T0604
R&B softening point	°C	50	T0606
Ductility at 10 °C	cm	49.4	T0605
Solubility	%	99.7	T0607
Flash point	°C	328	T0611
Density at 15 °C	g/cm^3^	1.040	T0603

**Table 2 materials-14-05379-t002:** Basic Properties of aggregate and mineral filler.

Materials	Basic Properties	Unit	Test Results	Test Method
Coarse Aggregate	Relative apparent density (R.A.D)	-	2.69	T0304
	Water absorption (W.A)	%	0.20	T0307
	Aggregate crushing value (C.V)	%	20.0	T0316
	Abrasion value (A.V)	%	26	T0317
	Flat and elongated particles	%	13	T0312
Fine Aggregate	Relative apparent density (R.A.D)	-	2.68	T0328
	Clay content	%	2.3	T0333
	Sand equivalent (S.E)	%	72.8	T0334
Mineral filler	Relative apparent density (R.A.D)	-	2.81	T0352
	Moisture content (M.C)	%	0.08	T0332

**Table 3 materials-14-05379-t003:** Test results of RAP material.

Materials	Physical Properties	Unit	Test Results	Test Method
Asphalt binder in RAP	Penetration at 25 °C	0.1 mm	15.1	T0604
	Ductility at 15 °C	cm	57	T0605
	Ductility at 25 °C	cm	106	T0605
	R&B softening point	°C	71.9	T0606
	Dosage	%	4.4	-
Coarse aggregate in RAP	Flat and elongated pieces	%	3.4	T0312
	Aggregate crushing value	%	11.57	T0316
Fine aggregate in RAP	Angularity	%	21.14	T0344

**Table 4 materials-14-05379-t004:** Gradation of dry powder mortar.

Components	Cement	Sand	Fly Ash	Heavy Calcium	Additive
Dosage (%)	35	30.25	10	15	9.75

**Table 5 materials-14-05379-t005:** Mixture ratios of the matrix asphalt mixture.

Sieve Size (mm)		Percentage Passing of New Aggregate/%				Percentage Passing of RAP/%				Powder Filler/%	Asphalt Content/%
		13.2	9.5	4.75	2.36	13.2	9.5	4.75	2.36		
RAP/%	0	100	36.9	12.0	3.50	-	-	-	-	3.50	3.50
RAP/%	15	100	36.8	11.9	3.56	100	34.2	8.3	0	3.56	3.56
RAP/%	30	100	36.7	11.7	3.68	100	33.9	8.0	0	3.68	3.68
RAP/%	45	100	36.5	11.5	3.81	100	33.6	7.6	0	3.81	3.81
RAP/%	60	100	36.2	11.2	3.97	100	33.2	7.2	0	3.97	3.97

**Table 6 materials-14-05379-t006:** Technical requirements of cement mortar.

Testing Indices	Fluidity/s	Dry Shrinkage/%	Compressive Strength/MPa		Flexural Strength/MPa	
			7d	28d	7d	28d
Technical requirements	10~12	<0.2	≥70	≥90	≥10	≥13

Note: d is the unit of curing age, it means days.

**Table 7 materials-14-05379-t007:** Fluidity test results of cement mortar with different water–cement ratio.

Water–Cement Ratio	0.20	0.21	0.22	0.23	0.24
Fluidity/s	14.1	12.8	11.4	10.6	10.1

**Table 8 materials-14-05379-t008:** Test results of cement mortar with different water–cement ratio.

Water–Cement Ratio	Curing Time/d	Compressive Strength/MPa	Flexural Strength/MPa	Dry Shrinkage/%
0.22	7	72.8	10.9	0.087
	28	91.8	13.5	0.138
0.23	7	71.8	10.7	0.094
	28	91.3	13.3	0.144
0.24	7	70.6	10.0	0.092
	28	89.2	12.6	0.148

**Table 9 materials-14-05379-t009:** Comparative sequences of warm mix recycled semi-flexible materials.

Groups	$\mathrm{RAP}\;\mathrm{Content}/\%\;x_{1}$	$\mathrm{Asphalt}\;\mathrm{Content}/\%\;x_{2}$	$\mathrm{Curing}\;\mathrm{Time}/d\;x_{3}$	Strain Magnitude/με $x_{4}$
1	0	3.50	3	150
2	15	3.56	3	150
3	30	3.68	3	150
4	45	3.81	3	150
5	60	3.97	3	150
6	0	3.50	7	200
7	15	3.56	7	200
8	30	3.68	7	200
9	45	3.81	7	200
10	60	3.97	7	200

**Table 10 materials-14-05379-t010:** Reference sequences of road performance of warm mix recycled semi-flexible materials.

Groups	$\mathrm{DS}\;(\mathrm{Cycles}/\mathrm{mm})\;x_{01}$	$\mathrm{Flexural}\;\mathrm{Tensile}\;\mathrm{Strength}\;(\mathrm{MPa})\;x_{02}$	Failure Strain $(\mu \varepsilon )\;x_{03}$	$\mathrm{MS}(\%)\;x_{04}$	$\mathrm{TRS}(\%)\;x_{05}$	$\mathrm{Fatigue}\;\mathrm{Life}\;(\mathrm{Times})\;x_{05}$
1	26,000	6.66	1617	105.81	90.82	41,983
2	19,500	5.91	1432	104.24	89.21	63,654
3	17,333	5.67	1389	103.15	88.38	80,256
4	15,600	6.06	1486	102.35	87.36	109,867
5	13,000	6.62	1644	100.32	87.06	128,769
6	39,000	6.39	1544	108.68	94.26	5253
7	26,000	5.73	1377	106.79	93.56	6762
8	19,500	5.69	1336	105.47	92.04	8147
9	17,727	5.92	1418	104.27	91.42	9786
10	15,600	6.47	1604	102.68	90.91	11,534

**Table 11 materials-14-05379-t011:** Grey correlation degree of influencing factors.

Correlation Degree	RAP Content	Asphalt Content	Curing Time	Strain Magnitude
DS	0.562	0.682	0.624	N/A
Flexural tensile strength	0.579	0.508	0.631	N/A
MS	0.592	0.617	0.656	N/A
TRS	0.589	0.609	0.658	N/A
Fatigue life	0.644	0.473	N/A	0.491

## Data Availability

Data Sharing is not applicable.

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
