# Peer review of "Study on the Road Performance of Foamed Warm-Mixed Reclaimed Semi-Flexible Asphalt Pavement Material"

_materials, 2021, doi:10.3390/ma14185379_

Round 1
Reviewer 1 Report
- The language of your manuscript is really a mess. If I would be the Editor I would reject it without sending it to reviewers. You mix the concepts of asphalt and pavement etc. You need to spend time fixing it and probably with help of some asphalt specialists with good technical English.
- You need to improve your topic. It is hard to understand what it is about. As far as I read, your manuscript is not about pavement, but about asphalt materials. The word asphalt doesn`t even appear in your manuscript title.
- Abstract looks like a bunch of unrelated sentences stucked togeter. It should begin with a short introduction, followed by objectives, and concluded with major conclusions.
- Your introduction needs more information about asphalt mixture performance. I missing some general information regarding the performance of asphalt mixtures such as doi.org/10.1080/14680629.2017.1283353 and doi.org/10.1080/14680629.2021.1908408. Don`t limit yourself to the papers about nanoindentation, show us a broader spectrum.
- After the introduction, I just skip the pages because it is very hard to understand your writing. I would love to see the version of this manuscript with improved English.
Reviewer 2 Report
The authors of the paper entitled: “Study on pavement performance of warm-mixed reclaimed semi-flexible material” propose improvement of road performances of pavement material.
The authors studied reclaimed asphalt pavement (RAP) materials, which consisted of asphalt, coarse aggregates, fine aggregates, and mineral filler. Currently, technologies based on reused of waste materials are appreciated and constantly improved.
Their work is quite interesting but manuscript is prepared carelessly.
- Why did the authors choose the RAP with diameters smaller than 4.75 mm. Does it have any practical significance?
- What is the Marshall Test? Please, describe its possibilities and applications.
- Did the authors use the usual temperatures when making warm-mixed reclaimed matrix asphalt mixtures?
- Here are no references.
- What did the authors use a cement mortar for?
- Please, explain carefully.
- Explanation of units is missing throughout the text, for example: 7d (in Table 6), 150 µƐ in line 294.
Regarding the editorial form, I think the Conclusions should be written in a more concise way, while some of the text should be inserted to Results and Discussion. Also, the references are very poor.
In my opinion, this manuscript requires major revision.
Round 2
Reviewer 1 Report
Thank you for addressing my comments.
Reviewer 2 Report
The manuscript looks much better in the current version. It will probably require many editorial corrections, but the important thing is that it is now more understandable for readers.